# Matairesinol Induces Mitochondrial Dysfunction and Exerts Synergistic Anticancer Effects with 5-Fluorouracil in Pancreatic Cancer Cells

**DOI:** 10.3390/md20080473

**Published:** 2022-07-25

**Authors:** Woonghee Lee, Gwonhwa Song, Hyocheol Bae

**Affiliations:** 1Department of Biotechnology, Institute of Animal Molecular Biotechnology, College of Life Sciences and Biotechnology, Korea University, Seoul 02841, Korea; cleverwhl@korea.ac.kr; 2Department of Oriental Biotechnology, College of Life Sciences, Kyung Hee University, Yongin 17104, Korea

**Keywords:** matairesinol, pancreatic cancer, mitochondria dysfunction, 5-fluorouracil, anticancer drugs

## Abstract

Pancreatic ductal adenocarcinoma (PDAC) is one of the most aggressive types of cancer and exhibits a devastating 5-year survival rate. The most recent procedure for the treatment of PDAC is a combination of several conventional chemotherapeutic agents, termed FOLFIRINOX, that includes irinotecan, leucovorin, oxaliplatin, and 5-fluorouracil (5-FU). However, ongoing treatment using these agents is challenging due to their severe side effects and limitations on the range of patients available for PDAC. Therefore, safer and more innovative anticancer agents must be developed. The anticarcinoma activity of matairesinol that can be extracted from seagrass has been reported in various types of cancer, including prostate, breast, cervical, and pancreatic cancer. However, the molecular mechanism of effective anticancer activity of matairesinol against pancreatic cancer remains unclear. In the present study, we confirmed the inhibition of cell proliferation and progression induced by matairesinol in representative human pancreatic cancer cell lines (MIA PaCa-2 and PANC-1). Additionally, matairesinol triggers apoptosis and causes mitochondrial impairment as evidenced by the depolarization of the mitochondrial membrane, disruption of calcium, and suppression of cell migration and related intracellular signaling pathways. Finally, matairesinol exerts a synergistic effect with 5-FU, a standard anticancer agent for PDAC. These results demonstrate the therapeutic potential of matairesinol in the treatment of PDAC.

## 1. Introduction

Pancreatic cancer (PC) is the most aggressive cancer with the lowest 5-year survival rate for all stages combined (only 11%) and is the third leading cause of cancer death in the United States [1]. Among the various types of PC, pancreatic ductal adenocarcinoma (PDAC), which originates in the ducts of the exocrine pancreas, is the most prevalent and accounts for 95% of cases [2]. In addition to the aggressive nature of PDAC, this cancer is highly metastatic and does not exhibit early warning symptoms. Additionally, tumor development occurs in a deep location, and there are technical deficiencies in regard to trustworthy screening tests, thus making it difficult for physicians to diagnose PDAC at early stages [3]. Due to these problematic characteristics of PDAC, many experts have predicted that PDAC may become the second leading cause of cancer-related deaths by 2030 [4]. Mutationally activated *KRAS* genes have been detected in nearly all PDAC. For this reason, numerous efforts have recently been made to develop effective therapies for PDAC that molecularly target *KRAS*; however, all have failed to date [5]. Instead, chemotherapeutic agents including irinotecan, gemcitabine, and 5-fluorouracil (5-FU), or more recently FOLFIRINOX, the combination of irinotecan, 5-FU, leucovorin, and oxaliplatin, have been commonly used for the treatment of PDAC patients [6]. However, a number of side effects associated with FOLFIRINOX have been reported, including diarrhea, neurotoxicity, fatigue, and myelosuppression [7,8]. Therefore, it is essential that novel therapeutic anticancer agents that are safer and more effective than conventional therapeutics must be developed.

Lignans are natural compounds possessing a diphenolic structure that are widely present in plants such as seeds, vegetables, and fruits [9]. Matairesinol is a dibenzylbutyrolactone lignan that can be extracted from the *Forsythia suspensa* fruit and the marine seagrass *Halophila stipulacea* [10,11]. The pharmacological and biological properties of matairesinol have been reported in various fields of medicine, and they include anti-allergic [10], neuroprotective [12], and anti-osteoporotic activities [13]. Moreover, there have been numerous reports that matairesinol exhibits anti-cancer activity in various types of cancer, including prostate, breast, cervical, and pancreatic cancer [14,15,16,17]. However, the detailed molecular mechanism underlying the anticancer effect of matairesinol in the context of pancreatic cancer has not yet been elucidated.

Therefore, in our study, we demonstrated the inhibitory effects of matairesinol on the progression of PANC-1 and MIA PaCa-2 pancreatic cancer cells. Our study aimed to (1) identify the intracellular alterations induced by matairesinol in terms of cell proliferation, apoptosis, the production of reactive oxygen species (ROS), mitochondrial function, homeostasis of calcium ions, and cell migration; (2) demonstrate the signaling pathway associated with cellular survival mediated by matairesinol in PC cells; and (3) verify the synergistic effect of matairesinol with conventional anticancer agents against PDAC and 5-FU.

## 2. Results

### 2.1. Matairesinol Inhibited Cell Proliferation and Progression in PC Cells

To verify the antiproliferative effect of matairesinol in the pancreatic ductal adenocarcinoma cells PANC-1 and MIA PaCa-2, both cell lines were treated with various concentrations of matairesinol for 48 h. The proliferation of both PC cell lines gradually decreased in response to increased concentrations of matairesinol. In particular, an 80 µM concentration of matairesinol inhibited proliferation by 48% in PANC-1 cells and 50% in MIA PaCa-2 cells (*p* < 0.001 for both cells) (Figure 1A,B). We set 80 µM as the optimal dose for matairesinol treatment due to its inhibitory effect on approximately half of the cells. Proliferating cell nuclear antigen (PCNA) is highly expressed in PDAC, which is closely involved in poor prognosis [18]. Based on the observation that matairesinol could suppress the growth of PC cells, the protein level of PCNA was further confirmed in response to matairesinol treatment through the use of Western blotting. The intensity of PCNA in the blot was decreased considerably in response to increased concentrations of matairesinol. Treatment with matairesinol (80 µM) reduced PCNA expression by 30% and 33% in both PC cell lines (*p* < 0.001 for both cell lines) (Figure 1C). Subsequently, we evaluated the inhibitory effect of matairesinol on spheroid formation in a three-dimensional (3D) environment using the hanging drop method. The relative total area of the formed spheroids was reduced in response to matairesinol treatment by 78% (*p* < 0.05) in PANC-1 cells and by 61% (*p* < 0.01) in MIA PaCa-2 cells (Figure 1D).

### 2.2. Matairesinol Intensified Apoptosis Induction and ROS Accumulation in PC Cells

Based on the result of antiproliferative activity of matairesinol against the PC cells, we investigated whether matairesinol could trigger apoptosis in PC cells by conducting an Annexin V and PI double staining assay. As matairesinol concentrations in PC cells were increased, late apoptotic cells also steadily increased in both PC cell lines. In particular, in response to 80 µM of matairesinol, the relative late apoptosis was increased by up to 196% in PANC-1 (Figure 2A) and by 261% in MIA PaCa-2 cells (Figure 2B) (*p* < 0.001 for both cells). We further investigated the apoptotic effect of matairesinol by confirming the level of BAX (a pro-apoptotic regulator) using Western blot analysis. In response to matairesinol, the expression of BAX was significantly increased in both PC cell lines compared to levels in the control (Figure 2C). Based on a report indicating that the accumulation of ROS could trigger apoptosis in various cancer cells [19], we examined whether matairesinol could promote the generation of ROS products in PC cells. Flow cytometry data revealed that after treatment with matairesinol, the relative ROS production increased in a dose-dependent manner in both PC cell lines (Figure 2D,E). In response to 80 µM matairesinol treatment, ROS production in PANC-1 cells was considerably increased to 447% and 548% in MIA PaCa-2 cells (*p* < 0.001 for both cells), and this is comparable to the positive control treatment with hydrogen peroxide. Collectively, our results indicated that matairesinol induced apoptosis in PC cells via oxidative stress.

### 2.3. Matairesinol Provokes Mitochondrial Dysfunction through MMP Loss and Calcium Dysregulation in PC Cells

As it is widely established that ROS production can contribute to mitochondrial dysfunction [20], we further investigated mitochondrial conditions with respect to mitochondrial membrane potential (MMP) and calcium regulation. First, we confirmed the depolarization of the mitochondrial membrane by measuring MMP. In both PC cell lines, the ratio of JC-1 green monomers to whole cells was markedly increased from 5.6% to 16.8% (*p* < 0.001) in PANC-1 cells and from 1.8% to 31.9% (*p* < 0.001) in MIA PaCa-2 cells in response to matairesinol treatment (Figure 3A,B). These increments were comparable to those of the positive control and valinomycin treatment in both PC cell lines. Next, we investigated cytosolic and mitochondrial calcium concentrations using Fluo-4 and Rhod-2 dyes. Flow cytometry data revealed that matairesinol gradually increased the cytosolic and mitochondrial calcium levels in both PC cell lines. Treatment with 80 µM matairesinol increased cytosolic calcium levels by 280% and 551% in both PC cell lines (*p* < 0.001 for both cell lines) (Figure 3C,D). Additionally, mitochondrial calcium levels were also significantly increased by 167% (*p* < 0.01) in PANC-1 cells and 192% (*p* < 0.001) in MIA PaCa-2 cells in response to the same dose of matairesinol (Figure 3E,F). Overall, our results indicated that matairesinol disturbed the physiological function of mitochondria through the loss of MMP and calcium dysregulation. 

### 2.4. Matairesinol Mitigates Migratory Ability in PC Cells

Invasion analysis and transwell migration assays have demonstrated that matairesinol suppresses cell invasion and migration, both of which are closely related to cell growth and metastasis of PDAC. In the invasion assay, the distances between two cell populations were significantly increased in response to treatment with matairesinol (80 µM) by up to 191% and 170% in both PC cell lines (*p* < 0.01 for both cells) compared to that of the control (Figure 4A). Moreover, according to the results of the transwell migration assay, migrated cells were somewhat reduced by treatment with matairesinol (80 µM) in both PC cells by 79% and 86% (*p* < 0.05) compared to that of the control (Figure 4B). Subsequently, the transcriptional levels of invasive genes related to cell migration were evaluated by qPCR. In PANC-1 cells, 80 µM matairesinol treatments slightly lowered the transcriptional levels of *vascular endothelial growth factor C* (*VEGFC*) and *forkhead box protein M1* (*FOXM1*) by approximately 0.7-fold for both genes. In contrast, in MIA PaCa-2 cells, the expression of these genes was significantly reduced by 49% (*p* < 0.01) and 47% (*p* < 0.05), respectively, in response to treatment with 80 µM matairesinol (Figure 4C,D). Moreover, the gene expression of *matrix metallopeptidase 1* (*MMP1*) was slightly reduced in PANC-1 cells but was significantly decreased by 37% (*p* < 0.01) in MIA PaCa-2 cells in response to matairesinol (80 µM) (Figure 4E). Furthermore, matairesinol treatment (80 µM) considerably increased the transcriptional expression of *plasminogen activator urokinase* (*PLAU*) in both PC cell lines (Figure 4F). These results revealed that matairesinol attenuated PC cell invasiveness.

### 2.5. Signaling Pathways Associated with Antitumor Effects of Matairesinol in PC Cells

Next, we evaluated the signaling pathways in PC cells that are associated with the antitumor effects of matairesinol by analyzing the phosphorylation levels of Ak strain transforming (AKT) and mitogen-activated protein kinase (MAPK). Matairesinol upregulated the phosphorylation of the proteins of interest, with the exception of ERK1/2, in both PC cell lines. The abundance of phosphorylated JNK increased in a dose-dependent manner in both of the PC cell lines. These levels were increased by 3.8-fold (*p* < 0.001) in PANC-1 cells and by 1.4-fold (*p* < 0.05) in MIA PaCa-2 cells (Figure 5A). Additionally, the phosphorylation levels of AKT were increased in response to matairesinol treatment (80 µM) by up to 1.9-fold in PANC-1 cells and by 1.8-fold in MIA PaCa-2 cells (*p* < 0.01 for both cells) (Figure 5B). Moreover, in response to matairesinol treatment (80 µM), phosphorylated P38 levels were 4.5-fold and 2.4-fold higher in PANC-1 and MIA PaCa-2 cells, respectively (*p* < 0.001 for both cells) (Figure 5C). However, phosphorylated ERK1/2 levels slightly increased in response to 20 µM treatment but decreased rapidly after treatment with 40 µM (Figure 5D). In response to 80 µM matairesinol, the phosphorylation levels of ERK1/2 were reduced by 0.49-fold in PANC-1 cells and by 0.55-fold in MIA PaCa-2 cells (*p* < 0.001 for both cells). These results suggest that matairesinol regulates the phosphorylation of the AKT and MAPK signaling pathways related to PC cell progression.

### 2.6. Synergistic Effects of Matairesinol with Anticancer Drugs in Respect to Mitochondrial Dysfunction

We investigated the synergistic effects of matairesinol when administered in combination with 5-FU, a standard anticancer drug, against PDAC. First, matairesinol (80 µM) combined with 5-FU (20 µM) exhibited a synergistic antiproliferative effect from 73% to 34% in PANC-1 cells and from 74% to 30% in MIA PaCa-2 cells compared to that from treatment with 5-FU alone (*p* < 0.001 for both cells) (Figure 6A). We further evaluated the synergistic effects of matairesinol and 5-FU on apoptotic cell death induced by ROS accumulation. In PANC-1 cells, the ratio of late apoptotic cells moderately increased in response to matairesinol (80 µM) combined with 5-FU when compared to that from 5-FU alone. In contrast, the relative apoptotic MIA PaCa-2 cells were significantly increased in response to co-treatment with matairesinol (80 µM) and 5-FU from 473% to 704% (*p* < 0.05) compared to that from treatment with 5-FU (Figure 6B). Additionally, the accumulation of ROS in both PC cell lines was significantly increased by the combination of matairesinol and 5-FU (Figure 6C). To confirm the synergistic effect of matairesinol and 5-FU on the regulation of mitochondrial function and calcium, we investigated the proportion of JC-1 monomers and the calcium concentration in the cytosol and mitochondria of PC cells. In MIA PaC-2 cells, the ratio of JC-1 monomers was significantly increased by matairesinol combined with 5-FU compared to that with 5-FU alone; however, there was no significant impact in PANC-1 cells (Figure 6D). Moreover, the relative cytosolic calcium concentration in both PC cells was significantly increased by a combination of matairesinol and 5-FU compared to that from 5-FU treatment alone from 243% to 405% (*p* < 0.01) in PANC-1 cells and from 138% to 238% (*p* < 0.001), respectively (Figure 6E). Furthermore, the relative mitochondrial calcium ion levels were also significantly increased after matairesinol treatment with 5-FU from 155% to 366% in PANC-1 cells and from 163% to 267% in MIA PaCa-2 cells (*p* < 0.001 for both cells) (Figure 6F). These results imply that the combined treatment with matairesinol and 5-FU exerted additional effects on the inhibition of cell proliferation, apoptotic cell death, and ROS production, all of which led to mitochondrial dysfunction and calcium dysregulation.

## 3. Discussion

As illustrated in Figure 7, matairesinol resulted in the inhibition of cellular growth, PCNA expression, and spheroid formation in PC cells and also caused apoptosis in PC cells. Additionally, matairesinol induced ROS accumulation, MMP reduction, and calcium influx. Moreover, matairesinol suppressed the expression of invasive genes and attenuated the migration of PC cells. Furthermore, matairesinol promoted MAPK and AKT signalling associated with the proliferation and progression of PC cells. Finally, 5-fluorouracil (5-FU), a chemotherapeutic agent prescribed for PDAC, exerted synergistic effects with matairesinol in regard to mitochondrial dysfunction and calcium dysregulation, thereby improving the efficiency of matairesinol for treating PC cells. Collectively, we elucidated the intracellular mechanisms of matairesinol that exhibit anticancer effects in PC cells.

The 5-year survival rate for PDAC is only 11%, primarily due to the observation that few chemotherapeutic agents have been demonstrated to be effective against PDAC patients. Gemcitabine is commonly used as a chemotherapeutic agent in clinics; however, the survival rates in response to this agent are still low. Moreover, treatment with 5-FU alone, which is commonly used for the treatment of colon cancer, resulted in no efficient therapeutic improvement in PDAC patients. Recently, FOLFIRINOX, a multidrug combination regimen, significantly improved the survival of patients with advanced PC. However, it cannot be administered at all stages of PDAC due to its cytotoxicity [21]. Over the years, matairesinol, a lignan present in seagrass, has been extensively studied due to its various biomedical properties that include anti-tumor, anti-inflammatory, and antioxidant effects [11,22]. Although matairesinol exhibits biological functions in various types of cancer, its intracellular mechanisms have not been studied in pancreatic cancer cells.

In previous cancer studies, matairesinol exhibited anti-tumor activity and triggered apoptosis in a prostate cancer model [14,23]. Additionally, matairesinol suppresses colon cancer cells by regulating Wnt/β-catenin signalling [24]. In agreement with these results, our experimental results explicitly suggest that matairesinol suppressed growth and triggered apoptosis induction in PC cells. First, reduced expression of proliferating cell nuclear antigen (PCNA), a marker for proliferating cells [25], was observed in response to matairesinol treatment in PC cells. Moreover, 3D cell aggregation was inhibited by matairesinol as demonstrated by spheroid formation, thus suggesting that matairesinol possesses the potential to suppress cell growth and tumor formation in PC cells. Second, apoptosis induced by matairesinol was further confirmed by the increased expression of the proapoptotic protein BAX in PC cells. Mitochondria play an important role in regulating the intrinsic apoptotic pathway [26]. Intrinsic apoptosis through mitochondria is accompanied by mitochondrial outer membrane permeabilization and the activation of proapoptotic proteins, including BAX and BAK [27]. BAX activation is a prerequisite for mitochondrial dysfunction and the mitochondrial program of apoptosis [28]. Following mitochondrial outer membrane permeabilization by BAX or BAK, an additional pathway leading to apoptosis occurs [29]. Although further studies regarding the comprehensive mechanisms of apoptosis induced by matairesinol in PC cells are required, it could lead to intrinsic apoptosis through the activation of BAX protein in the mitochondria of PC cells.

Our present study also suggests that matairesinol can lower the mitochondrial membrane potential (MMP) in PC cells. Alterations in the properties of the mitochondrial membrane can be a signal for apoptosis, among which changes in MMP are crucial indicators of cell survival [30]. Additionally, we demonstrated that matairesinol triggered the disruption of calcium homeostasis in PC cells by increasing both mitochondrial and cytosolic calcium ions. Calcium in the context of cancer is a key regulator of intracellular processes, including ROS production and cell progression [31,32]. Excessive increases in calcium can alter the characteristics of respiratory chain complexes in mitochondria, and this triggers mitochondrial ROS generation [33]. Mitochondrial calcium overload leads to the swelling of mitochondria, and this causes damage to the outer membrane and, subsequently, the emission of apoptotic proteins from the mitochondria into the cytosol [34]. Moreover, it has been reported that mitochondrial dysfunction can cause calcium influx into the cytosol and lead to apoptosis in cancer cells [35]. Although the mutual interplay between ROS and calcium signaling induced by matairesinol in PC cells requires further study, it may induce mitochondrial dysfunction via MMP alteration, ROS production, and calcium dysregulation in PC cells.

Recent studies have suggested that the epithelial–mesenchymal transition (EMT), where epithelial cells lose junctional complexes and acquire a motile phenotype, is closely implicated in the expression of malignant properties in metastatic cancer and the initiation of metastasis [36]. These changes from the epithelial to mesenchymal phenotype are characterized by the ability to acquire invasiveness, migration, and resistance to apoptosis [37]. There have been some reports that several genes related to EMT in PDAC, including *forkhead box protein M1* (*FOXM1*), *matrix metalloproteinase-1* (*MMP1*), and *vascular endothelial growth factor C* (*VEGFC*), are targets for the treatment of PDAC. For example, the acquisition of EMT in PDAC is strongly associated with the expression of *FOXM1*, which functions as a regulator of Snail and stimulates EMT [38]. A recent study reported that *MMP1* promotes the metastasis of PC and that the inhibitory regulation of *MMP1* with endogenous microRNA can attenuate the metastatic ability of pancreatic cancer [39]. Additionally, the suppression of *VEGFC* expression can effectively inhibit PDAC [40]. Furthermore, *PLAU* that is upregulated in PDAC and known to promote EMT progression can provide a target for the treatment of PDAC by inhibiting its expression [41] The results of our migration and invasion assays indicated that matairesinol attenuated the invasiveness of PC cells. Moreover, the present qPCR data further confirmed the reduced expression of the invasive genes *FOXM1*, *MMP1*, *VEGFA*, and *PLAU* following matairesinol treatment. Collectively, our results revealed that matairesinol suppressed cell migration through the regulation of invasive genes in PC cells.

The intracellular signalling pathway is important for the regulation of cancer cells and can often provide a therapeutic target for cancer treatment. There have been many efforts to develop PDAC anticancer drugs targeting *KRAS*, which is permanently activated in PDAC; however, they have all failed [42,43]. Inhibitors of RAF-MEK-ERK, the downstream target of KRAS and PI3K-AKT-mTOR signaling, have also recently been mentioned as candidates for PDAC treatment [44,45]. These inhibitors have demonstrated potent synergistic effects in a mouse model [42]. Moreover, it has been reported that direct inhibition of ERK through the ERK 1/2-specific pharmacologic inhibitor (SCH772984) effectively suppressed KRAS-mutant PDAC, simultaneously elevating the phosphorylation of AKT [46]. Furthermore, recent studies have shown that the inhibition of ERK or MEK using a pharmacological inhibitor against PDAC cells promotes protective effects via the activation of autophagy [47,48]. In the present study, matairesinol regulated the MAPK signaling pathway and AKT. Matairesinol also elevated the phosphorylation levels of JNK, P38, and AKT but suppressed the phosphorylation of ERK1/2 in PC cells. However, we could not determine how the proteins involved in MAPK signaling and AKT regulate the progression and survival of PC cells, both of which were suppressed by matairesinol. Therefore, further studies are required.

It has been reported that matairesinol exerts a synergistic anticancer effect in cancer in combination with other lignans [49,50]; however, there have been no reports of synergistic effects with conventional standard anticancer drugs such as 5-FU. 5-FU is typically used for the treatment of several types of cancers, including colorectal, breast, and pancreatic cancers. However, 5-FU treatment alone is not selected to treat PDAC due to its lower effectiveness in clinical benefits [51]. Instead, in the current treatment of PDAC with 5-FU and FOLFIRINOX, a modified adjuvant regimen is typically used at the cost of increased toxicity. Although FOLFIRINOX is effective against PDAC, chemoresistance and cytotoxicity have become concerns [52,53]. In the present study, we demonstrated the synergistic effects of matairesinol with the conventional antidrug 5-FU in PDAC. Our data indicated that matairesinol exerts synergistic effects with 5-FU in regard to the inhibition of proliferation, induction of apoptosis, ROS production, MMP loss, and calcium regulation in PC. Moreover, the anticancer effects of matairesinol against PC cells, including anti-proliferation and calcium regulation, were superior to those of 5-FU. Therefore, matairesinol can be used as a potent chemotherapeutic agent and can be applied to improve the anticancer effects of conventional anticancer drugs such as 5-FU in PDAC. However, our limitation is that we did not compare the mode of action of matairesinol against normal pancreatic cells and did not confirm the effect of matairesinol on pancreatic cancer in animal models. Therefore, further studies should be conducted on the mode of action of matairesinol in normal pancreatic cells and the effect of matairesinol in an in vivo model of pancreatic cancer.

## 4. Materials and Methods

### 4.1. Chemicals and Antibodies

Matairesinol and 5-fluorouracil (5-FU) were both purchased from Sigma-Aldrich (St. Louis, MO, USA). Both chemicals were dissolved in dimethyl sulfoxide (DMSO). Information regarding the antibodies used in our study is presented in Table 1.

### 4.2. Cell Culture

The human pancreatic ductal adenocarcinoma cell lines PANC-1 and MIA PaCa-2 were obtained from the Korean Cell Line Bank (Seoul, Korea). For the maintenance of both cell types, the cells were cultured in DMEM containing 10% fetal bovine serum and 1% penicillin-streptomycin. After the cells were seeded into 6-well or 96-well plates, they were maintained until they reached 70% confluence. The cells were then starved in serum-free DMEM overnight and treated with various concentrations of matairesinol with or without 5-FU. All experiments were performed in triplicate.

### 4.3. Cell Proliferation Analysis

We confirmed the proliferation of PC cells using Cell Proliferation Kit I (Roche, Basel, Switzerland). The PC cells seeded into 96-well plates were treated with different doses of matairesinol for 48 h and then incubated in 10 µL of MTT tetrazolium salt at 37 °C for 4 h. Subsequently, 100 µL of solubilization buffer was added, and the cells were incubated at 37 °C overnight in dark. The optical density was measured at 560 and 650 nm using a microplate reader.

### 4.4. Spheroids Formation Analysis

For the formation of cell spheroids, PC cells were maintained using the hanging drop method for three days with matairesinol (80 μM). The spheroid morphology was detected using a DM3000 microscope (Leica Microsystems GmbH, Wetzlar, Germany). Whole images of each spheroid were transferred to ImageJ (NIH, Bethesda, MD, USA), and the total area of PC cell aggregation was quantified as previously described [54].

### 4.5. Reactive Oxygen Species (ROS) Assay

The intracellular production of ROS was detected using 2,7-dichlorofluorescin diacetate (DCFH-DA, Sigma-Aldrich), where ROS production converts fluorescent 2,7-dichlorofluorescin (DCF). Briefly, PC cells were incubated with matairesinol (0, 20, 40, and 80 μM) with or without 5-FU for 1 h. Subsequently, treated PC cells were incubated with DCFH-DA for 30 min and then harvested. The fluorescent signals from DCF were detected using a flow cytometer.

### 4.6. Mitochondrial Membrane Potential (MMP) Evaluation

Mitochondrial membrane depolarization was analyzed using a mitochondrial staining reagent (Sigma-Aldrich). Briefly, PC cells were treated with matairesinol (0, 20, 40, and 80 μM) with or without 5-FU for 48 h. PC cells were stained with JC-1 dye for 20 min and then harvested. JC-1 fluorescence intensity was estimated using a flow cytometer.

### 4.7. Apoptosis Analysis

Annexin V and propidium iodide (PI) double staining was used for apoptosis analysis using an apoptosis detection kit I (BD Biosciences, Franklin Lakes, NJ, USA). Briefly, after treatment with matairesinol or matairesinol combined with 5-FU for 48 h, PC cells were harvested and stained with Annexin V and PI dyes for 15 min at room temperature. The fluorescence intensity was detected using a flow cytometer.

### 4.8. Mitochondrial Calcium Measurement with Rhod-2

The mitochondrial calcium concentration in PC cells was measured using the fluorescent dye Rhod-2 (Invitrogen, Carlsbad, CA, USA). Briefly, the PC cells were incubated with various doses of matairesinol (0, 20, 40, and 80 μM) with or without 5-FU for 48 h. Then, the cells were harvested and loaded with Rhod-2 at 4 °C for 30 min. The emission intensity of the fluorescent Rhod-2 was detected using a flow cytometer.

### 4.9. Intracellular Calcium Measurement with Fluo-4

The intracellular calcium concentration in PC cells was measured using the fluorescent dye Fluo-4 (Invitrogen). Briefly, the PC cells were incubated with various doses of matairesinol (0, 20, 40, and 80 μM) with or without 5-FU for 48 h. Then, the cells were harvested and loaded with Fluo-4 at 37 °C for 20 min. The fluorescence emission intensity of Fluo-4 was measured using a flow cytometer.

### 4.10. Cell Migration Using a Transwell Assay

PC cells treated with matairesinol (80 μM) were seeded onto SPLInsert^TM^ hanging membranes (SPL Life Sciences, Pocheon, Korea). After incubation for 24 h, the PC cells in the inserts were rinsed several times with PBS, fixed with methanol for 10 min, and stained with hematoxylin for 1 h. After the membranes were rinsed twice with PBS, the cells on the membrane inside the insert were removed using a cotton swab. The membranes were detached from the inserts, placed on glass slides, and covered with the Permount solution. The migrated cells were counted under a DM3000 microscope.

### 4.11. Cell Invasion Analysis

The invasiveness of PC cells was assessed using 35-mm culture dishes (Ibidi, Munich, Germany) in accordance with the manufacturer’s instructions. PC cells were seeded onto each dish and maintained at 90% confluency. The PC cells were serum-starved for 16 h and treated with matairesinol (80 µM) after the wall positioned in the middle part of the dish was removed. Images of the gap between the two clusters on both sides were acquired using a DM3000 microscope. The invasiveness of PC cells was quantified according to the gap distance using ImageJ software.

### 4.12. Quantitative PCR (qPCR)

Total RNA was extracted from PC cells using TransZol Up reagent (TransGen Biotech, Beijing, China) according to the manufacturer’s instructions, and the concentration of the extracted total RNA was determined using a spectrophotometer (Thermo Fisher Scientific Inc., Waltham, MA, USA). Complementary DNA was synthesized using 2000 ng of RNA with AccuPower^®^ RT PreMix (Bioneer, Daejeon, Korea), and products of interest were amplified by qPCR using SYBR Green and the CFX Connect Real-Time System (Bio-Rad Laboratories Inc., Hercules, CA, USA) under the following temperature conditions: 95 °C for 3 min, followed by 40 cycles at 95 °C for 20 s, 64 °C for 40 s, and 72 °C for 1 min. We confirmed that only one product was amplified using a melting curve from 55 to 95 °C. The relative mRNA expression levels were quantified using the $2^{-\Delta \Delta \mathrm{CT}}$ method. GAPDH expression was used to normalize the gene expression. The primers used in this study are listed in Table 2.

### 4.13. Western Blot


Proteins were isolated from PC cells using RIPA lysis buffer (Sigma-Aldrich), quantified using the Bradford reagent (Bio-Rad Laboratories Inc.), and resolved by SDS-PAGE. The proteins were transferred onto polyvinylidene fluoride (PVDF) membranes and incubated with primary antibodies at 4 °C for 16 h and then with secondary antibodies for 1 h. Target proteins were detected using West-Q Pico Chemiluminescent Substrate (GenDEPOT, Katy, TX, USA) and an Alliance Mini HD9 acquisition system (Alliance UVItec Ltd., Cambridge, UK).

### 4.14. Synergistic Drug Combination Analysis

The synergistic effects between matairesinol and 5-FU were evaluated using the Bliss independence model [55]. Briefly, the synergistic effects between the two drugs were analyzed by the difference (Δ) between the observed (O) and expected (E) inhibitory rates of the combined treatment. E is determined as follows: E = X + Y − XY, where X and Y are the relative inhibitory rates of a single treatment. The positive Δ represents a synergistic effect and the negative Δ represents an antagonistic effect between two drugs. The Bliss values for matairesinol (80 μM) with 5-FU (20 μM) were determined based on our results for the proliferation analysis.

### 4.15. Statistical Analysis

All data were assessed using analysis of variance (ANOVA), followed by Dunnett’s post hoc test using the Statistical Analysis System (SAS, Cary, NC, USA). Statistical significance was set at *p* < 0.05. Data are presented as mean ± standard deviation.

## 5. Conclusions

In conclusion, we elucidated the mechanisms underlying the antitumor activity of matairesinol in PC cells. We confirmed that matairesinol suppressed cell progression and migration, triggered apoptosis and mitochondrial dysfunction through MMP loss, and disturbed calcium regulation. Additionally, these effects were facilitated in combination with 5-FU. Taken together, our data indicate that matairesinol may be an innovative therapeutic agent against PDAC. Our study is the first to verify that matairesinol exhibits synergistic effects with conventional chemotherapeutic agents in cancer cells. Although this study was limited to in vitro findings, these results could provide a basis for the development of novel chemotherapeutic agents and in vivo study.

## Figures and Tables

**Figure 1 marinedrugs-20-00473-f001:**
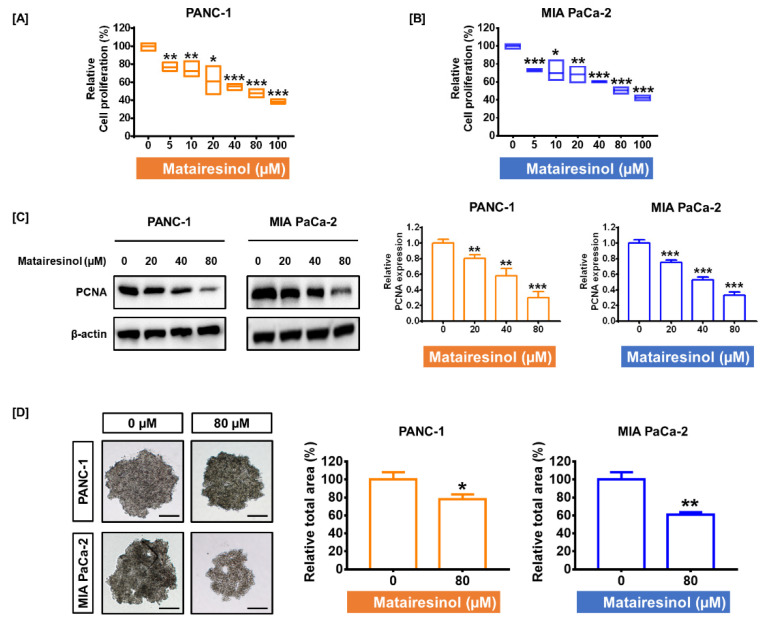
Antiproliferative effects of matairesinol on human pancreatic cancer cells. (**A**,**B**) Cell proliferation analysis was conducted in response to different matairesinol treatments (0, 5, 10, 20, 40, 80, and 100 µM) in PANC-1 and MIA PaCa-2. Relative cell proliferation gradually decreased with increasing matairesinol concentrations. (**C**) Immunoblot images of PCNA after treatment with matairesinol (0, 20, 40, and 80 µM) in PC cells. The relative PCNA expression was reduced in a dose-dependent manner in both PC cell lines as indicated in the bar graph. (**D**) The microscope images of spheroid formation in PC cells without and with matairesinol treatment (80 µM) using the hanging drop method. The relative total area of each spheroid is indicated in the bar graph. Spheroid formation in cells without and with matairesinol treatment. Scale bar, 100 µm. All experiments were conducted in triplicate. The degree of statistical significance among control and matairesinol-treated groups is represented by asterisks as determined using one-way analysis of variance (ANOVA), followed by Dunnett’s post hoc analysis (* *p* < 0.05, ** *p* < 0.01, and *** *p* < 0.001).

**Figure 2 marinedrugs-20-00473-f002:**
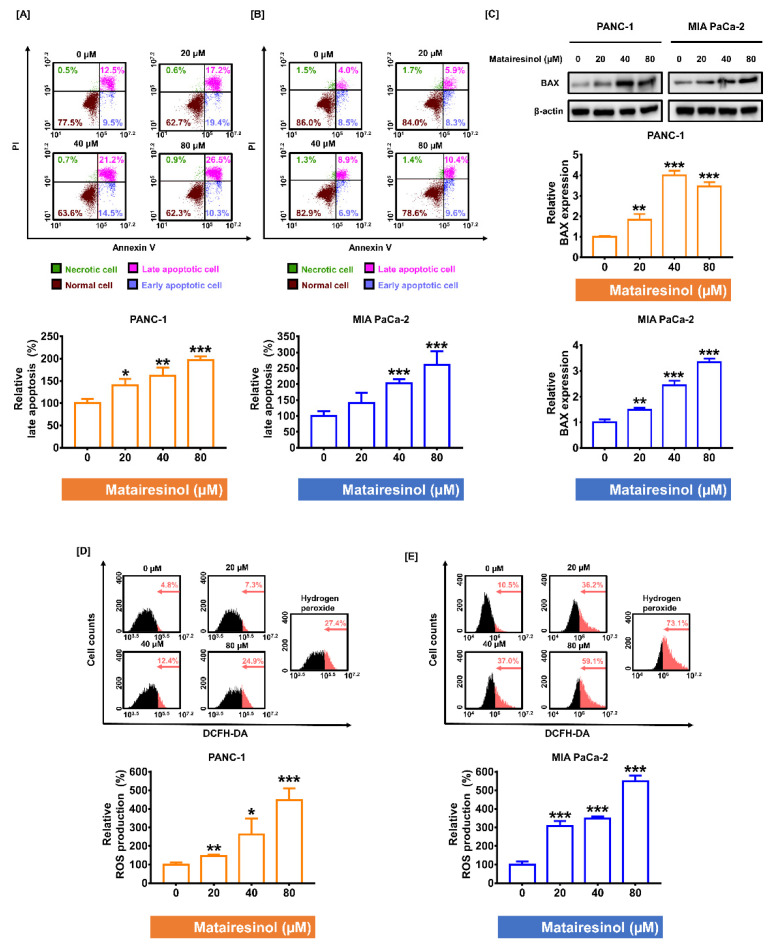
Dose-dependent anticancer impacts of matairesinol on apoptosis and ROS accumulation in PC cells. (**A**,**B**) Apoptotic PANC-1 and MIA PaCa-2 cells were analyzed using Annexin V and PI double staining assay in response to matairesinol treatment (0, 20, 40, 80 µM). (**C**) Immunoblot images of BAX after treatment with matairesinol (0, 20, 40, 80 µM) in both PC cell lines. (**D**,**E**) Intracellular ROS accumulation was evaluated as DCF fluorescence emission using flow cytometry. The treatment with H_2_O_2_ is the positive control group. The parts of red color in histogram indicate the relative ROS productions. All experiments were conducted in triplicate. The degree of statistical significance among control and matairesinol-treated groups is represented by asterisks, as determined using one-way analysis of variance (ANOVA), followed by Dunnett’s post hoc analysis (* *p* < 0.05, ** *p* < 0.01, and *** *p* < 0.001).

**Figure 3 marinedrugs-20-00473-f003:**
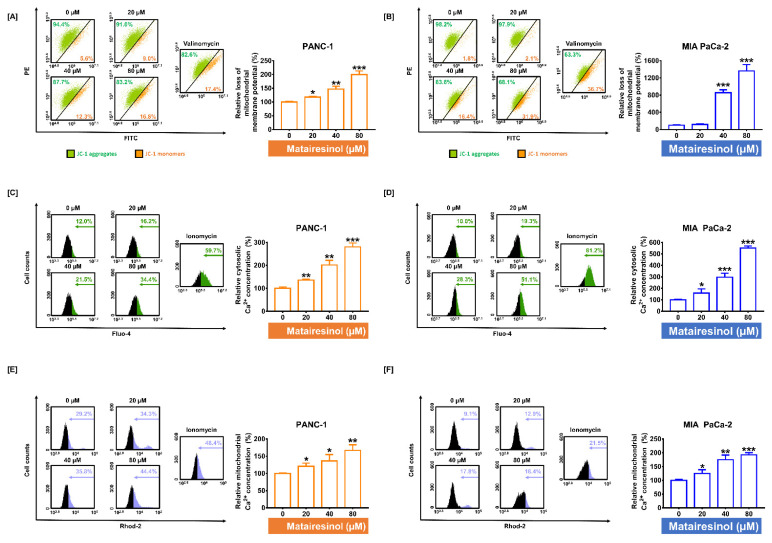
Changes in mitochondrial membrane potential (MMP) and calcium concentration in the cytosol and mitochondria of PC cells in response to matairesinol treatment (**A**,**B**) The effect of matairesinol on lowering of MMP was analyzed by JC-1 assay using flow cytometry. The proportion of JC-1 monomers is indicated in a bar graph as an indicator for loss of MMP. Valinomycin was used as a positive control. (**C**,**D**) Alterations in cytosolic calcium concentration were measured according to Fluo-4 fluorescence emission using flow cytometry. The parts of green color in histogram indicate the relative cytosolic calcium concentration. (**E**,**F**) Mitochondrial calcium concentration was evaluated as Rhod-2 fluorescence emission using flow cytometry with ionomycin as a positive control. The parts of blue color in histogram indicate the relative mitochondrial calcium concentration. Each assay was performed in triplicate. The degree of statistical significance among control and matairesinol-treated groups is represented by asterisks, as determined using one-way analysis of variance (ANOVA), followed by Dunnett’s post hoc analysis (* *p* < 0.05, ** *p* < 0.01, and *** *p* < 0.001).

**Figure 4 marinedrugs-20-00473-f004:**
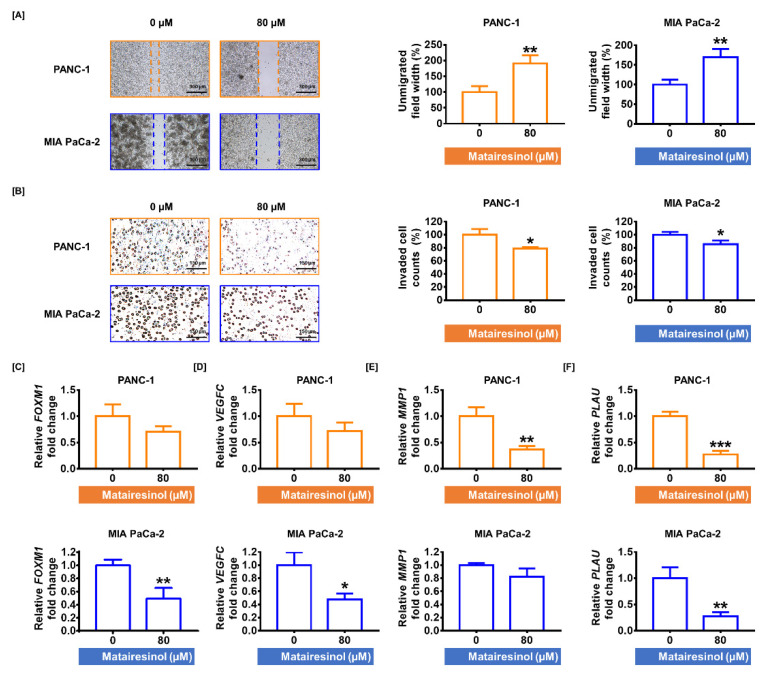
Matairesinol exerts an inhibitory effect on cell invasion and migration in PC cells. (**A**) Cell invasiveness was evaluated using Ibidi 35 mm culture dishes. The relative interspace distance between both sides of the cell clusters is represented in a bar graph as an indicator of cell invasiveness. (**B**) Transwell cell migration assays were conducted to analyze the migratory ability in response to matairesinol treatment in PC cells. (**C**–**F**) The transcriptional expression levels of (**C**) *FOXM1*, (**D**) *VEGFC*, (**E**) *MMP1*, (**F**) and *PLAU*, all of which are associated with cell migration in PC cells, were determined by qPCR. All experiments were conducted in triplicate. The degree of statistical significance among control and matairesinol-treated groups is represented by asterisks, as determined using one-way analysis of variance (ANOVA), followed by Dunnett’s post hoc analysis (* *p* < 0.05, ** *p* < 0.01, and *** *p* < 0.001).

**Figure 5 marinedrugs-20-00473-f005:**
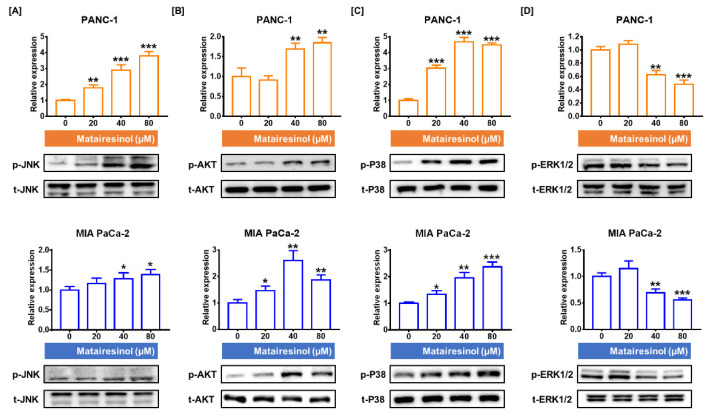
Relative phosphorylation levels of various proteins involved in the MAPK and PI3K signaling pathway in PC cells. (**A**–**D**) The intensities of phosphorylated (**A**) JNK, (**B**) AKT, (**C**) P38, and (**D**) ERK1/2 were estimated by immunoblotting tests. The protein level of each target was normalized according to the levels of each total protein. All experiments were conducted in triplicate. The degree of statistical significance among control and matairesinol-treated groups is represented by asterisks, as determined using one-way analysis of variance (ANOVA), followed by Dunnett’s post hoc analysis (* *p* < 0.05, ** *p* < 0.01, and *** *p* < 0.001).

**Figure 6 marinedrugs-20-00473-f006:**
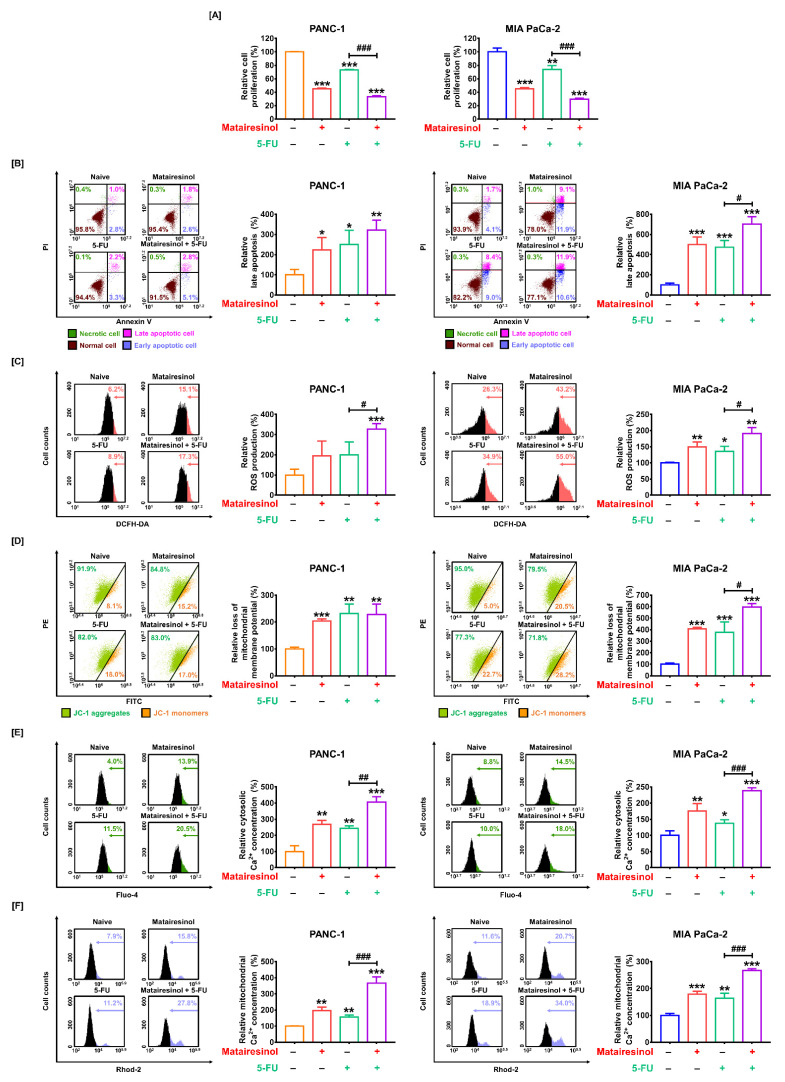
Complementary effects of matairesinol synergized with 5-FU in PC cells. (**A**–**F**) Complementary effects of matairesinol (80 µM) with 5-FU (20 µM) in (**A**) inhibition of proliferation, (**B**) apoptosis, (**C**) ROS production, (**D**) MMP loss, (**E**) cytosolic calcium concentration, and (**F**) mitochondrial calcium concentration. All experiments were conducted in triplicate. The degree of statistical significance among control and treatment groups is represented by asterisks (* *p* < 0.05, ** *p* < 0.01 and *** *p* < 0.001) and between the 5-FU only treatment group and the co-treatment group by crosshatches as determined using two-way analysis of variance (ANOVA), followed by Dunnett’s post hoc analysis (# *p* < 0.05, ## *p* < 0.01 and ### *p* < 0.001).

**Figure 7 marinedrugs-20-00473-f007:**
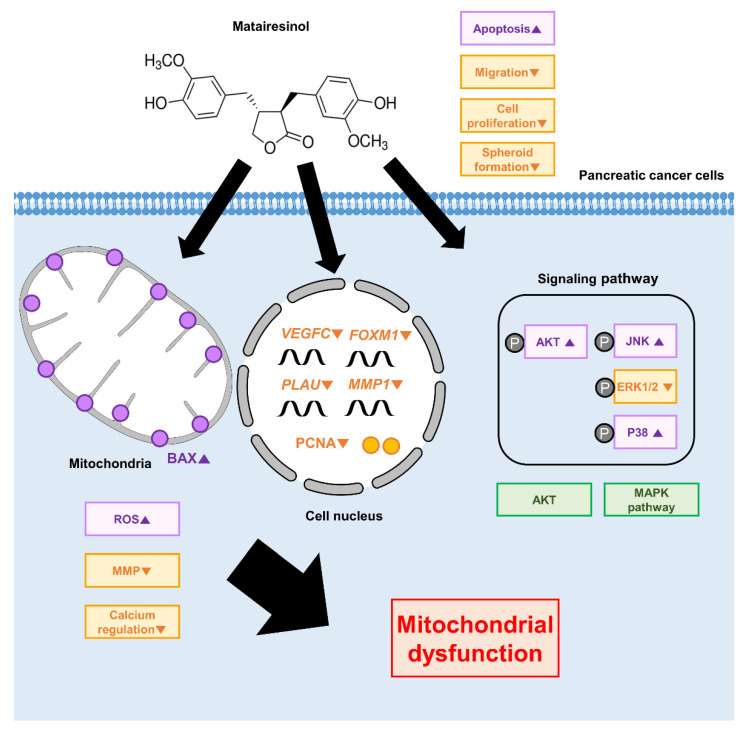
Diagrammatic illustration of the intracellular mechanism of matairesinol in pancreatic cancer cells.

**Table 1 marinedrugs-20-00473-t001:** The detailed information about antibodies we used.

Antibody	Catalog Number	Supplier	Dilution
PCNA	10205-2-AP	Proteintech	1:2000
BAX	50599-2-Ig	Proteintech	1:2000
p-JNK (Thr183/Tyr185)	4668	Cell Signaling Technology	1:1000
JNK	9252	Cell Signaling Technology	1:1000
p-ERK1/2 (Thr202/Tyr204)	9101	Cell Signaling Technology	1:1000
ERK1/2	4695	Cell Signaling Technology	1:1000
p-P38 (Thr180/Tyr182)	4511	Cell Signaling Technology	1:1000
P38	9212	Cell Signaling Technology	1:1000
p-AKT (Ser473)	4060	Cell Signaling Technology	1:1000
AKT	9272	Cell Signaling Technology	1:1000
β-actin	sc-47778	Santa Cruz Biotechnology	1:1000

**Table 2 marinedrugs-20-00473-t002:** The primers we used in qPCR.

Gene	Size (bp)	GenBank Accession No.	Primer Sequence (5′ → 3′)
*forkhead box protein M1 (FOXM1)*	104	NM_001243088.2	F: AGTCACACCCTAGCCACTGC
			R: ACCATTGCCTTTGTTGTTCC
*matrix metallopeptidase 1 (MMP1)*	147	NM_001145938.2	F: GGGAGCAAACACATCTGACC
			R: CTGCTTGACCCTCAGAGACC
*plasminogen activator, urokinase (PLAU)*	139	NM_002658.6	F: TGTGAGATCACTGGCTTTGG
			R: TTTTGGTGGTGACTTCAGAG
*vascular endothelial growth factor C (VEGFC)*	116	NM_005429.5	F: AGTTCCACCACCAAACATGC
			R: CCAATATGAAGGGACACAACG
*glyceraldehyde-3-phosphate dehydrogenase (GAPDH)*	149	NM_001256799.3	F: GGCTCTCCAGAACATCATCC
			R: TTTCTAGACGGCAGGTCAGG

## Data Availability

Data are contained within the article.

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
