# Peer review of "Matairesinol Induces Mitochondrial Dysfunction and Exerts Synergistic Anticancer Effects with 5-Fluorouracil in Pancreatic Cancer Cells"

_marinedrugs, 2022, doi:10.3390/md20080473_

Round 1
Reviewer 1 Report
Authors in their experimental study have mechanistically elucidated the anticancer effects of Matairesinol against pancreatic cancer cells. Study is of interest and well performed but lacks proper orientation of the results. I have raised few issues related to the manuscript, I hope authors could work on the comments to make it more interesting:
Major Comments:
1. In the abstract, authors stated in line 21-22 "However, it remains unclear if matairesinol exhibits effective anticancer activity against pancreatic cancer". This entire line does not holds true as previous study performed by Hang Chang et al 2017 have shown anti-cancer efficacy of these Lignons against pancreatic cancer cells. Please remove or modify the wrong statement. Do cite the reference Hang Chang 2017.
2. In the abstract, it is confusing to know that authors are stating matairesinol as a novel anticancer therapy, however, the overall manuscript suggest it as an adjuvant therapy in combination with 5-FU. Authors need to be sure about their saying; hence a better conclusive sentence is required.
3. In the introduction section, Lines 61-62 ". However, the anticancer effect of matairesinol in the context of pancreatic cancer has not yet been elucidated." This line is wrong and misleading. Hang Chang et al 2017 have shown anti-cancer efficacy of these Lignons against pancreatic cancer cells. Please correct the sentence and include this reference in the text. Interestingly, this study lacks any mechanistic aspect which could be deemed as an asset to present manuscript.
4. At the end of the Introduction section and Result section (2.6), how can authors be sure that Matairesinol exhibit synergistic effect but not an additive effect? More specifically authors should calculate synergistic or additive activity by calculating ISOBOLOGRAM etc. For more information read Ruo-Yue Huang et al 2019.
5. In the entire result section and figure legends, authors have not stated how many times each experiment was replicated. This is misleading and such missing information might ignite post-publication problems. Authors are requested to state how many times each individual experiment was replicated in the respective figure legend.
6. In the result section 2.1., authors stated that "Based on the observation that matairesinol could suppress the growth of PC cells, the protein level of PCNA was further confirmed in response to matairesinol treatment". I could not understand why suddenly PCNA levels were assessed. It could make readers confused. Authors need to describe why they only checked PCNA? Please do the needful.
7. In the result section 2.2., I could not understand the lines 105-106. Please reframe the lines to get the exact meaning.
8. In the result section 2.2., authors stated in lines 108-109 "In particular, in response to 80 µM of matairesinol the relative late apoptosis was increased by up to 196% in PANC-1 (Figure 2A) and by 261% in MIA PaCa-2 cells (Figure 2B)". I could not understand. How late apoptosis is calculated as 196% & 261% in the bar graph, whereas the FACS based dot plot shows late apoptosis as 26.5% & 10.4%? Authors please explain or simplify to what is shown in FACS data. At the moment bar graph are highly misleading.
9. In the figure 3A, 3B, 3C, 3D, FACS data in % does not correspond to the bar graph. It will be highly appreciable if authors could maintain uniformity between the FACS data and bar graph data.
10. In section 2.3., line 144-145, authors stated "Treatment with 80 µM matairesinol increased cytosolic calcium levels by 280% and 551% in both PC cell lines" (Figure 3C, 3D). Again there is same problem. How authors are calculating 280 or 551%? Whereas, FACS data limit cut-off is only 100%.
11. In section 2.4., authors have not justified the need for assessing only VEGF or FOXM1 or PLAU. Please specify the importance of assessing only these genes.
12. In section 2.6., title states "Synergistic effects of matairesinol….. ". However, the description states " First, matairesinol (80 µM) combined with 5-FU (20 µM) exhibited an additive antiproliferative effect ", This is confusing. Either it should be synergistic or it should be additive. Therefore, authors need to ascertain synergistic or additive activity by using ISOBOLOGRAM analysis. For more information read Ruo-Yue Huang et al 2019.
13. Bar graphs in Figure 6B-6F needs to be corrected based on the points raised in comment 8 and 9. Please do correct the bar graph in matching with FACS data.
14. Figure 7 needs to be corrected based on the ISOBOLOGRAM analysis, which will suggest Synergistic or Additive effect. Alternatively, authors could remove Big Red Arrow pointing inside the cell towards PCNA along with 5-FU structure. Since the soul of the manuscript is based on the importance of matairseinol, I would suggest to only depicts the mechanism of matairesinol.
15. In the figure 7, what does CELL PROGRESSION in rectangular box means? Does it means cell proliferation or else? Please correct.
16 . In the figure 7, Why only PCNA is in nucleus and not other genes (VEGFC, MMP1, FOXM1, PLAU). Manuscript does not show these genes or proteins to shuttle out of nucleus. So, nucleus size should be enhanced and encircling all these 4 genes.
17. In the figure 7, events such as Apoptosis, migration, cell proliferation should be shifted out of the cell structure as all these events are extracellular events.
18. Lines 328-338, needs to be supplement with more references, since few drugs such as Moxifloxacin and ciprofloxacin has shown their efficacy against pancreatic cancer cells by enhancing pERK levels and decreasing pAKT levels (Yadav V et al 2015). More such examples are asiatic acid, pemetrexed, lauryl gallat Taxol etc.
19. Section 4.11., authors did not state the method or software to quantify the gap distance for invasion analysis. Please mention.
20. Since in all the figures authors have used one-way ANOVA, it is important to mention the posthoc multiple comparison test. Please do the needfull for each figure legend.
Minor Comments:
1. In the abstract, authors stated in line 29-30 "Therefore, these results demonstrate ……context of PDAC as a novel anticancer therapy". This sentence is grammatically incorrect. Please reframe.
2. In section 2.6., Line 216-217 "................... and standard anticancer drugs against PDAC and 5-FU" are grammatically incorrect. Please rectify.
3. In the figure 7., small triangles showing the up or down regulation should also be placed for AKT and MAPK pathway proteins.
4. In the figure 7., decrease in the spheroid formation by matairesinol is missing. Since it is an extracellular event, it should be outside the cell structure.
5. Please correct the line 275-276 of discussion section. I could not understand.
6. In the section 4.12., please mention the amount of RNA (micro or nanogram) used for cDNA synthesis.
Author Response
[Comments and Suggestions for Authors]
[Reviewer 1]
Authors in their experimental study have mechanistically elucidated the anticancer effects of Matairesinol against pancreatic cancer cells. Study is of interest and well performed but lacks proper orientation of the results. I have raised few issues related to the manuscript, I hope authors could work on the comments to make it more interesting:
Response: We appreciate the reviewer’s valuable comments and suggestions regarding our manuscript. We have revised our manuscript according to the reviewers’ remarks and highlighted the changes in the manuscript's text in yellow.
Major Comments:
- In the abstract, authors stated in line 21-22 "However, it remains unclear if matairesinol exhibits effective anticancer activity against pancreatic cancer". This entire line does not holds true as previous study performed by Hang Chang et al 2017 have shown anti-cancer efficacy of these Lignons against pancreatic cancer cells. Please remove or modify the wrong statement. Do cite the reference Hang Chang 2017.
Response: We appreciate and agree with the reviewer’s valuable comments. As the reviewer mentioned, matairesinol had already shown anti-cancer efficacy against pancreatic cancer cells, as described by Hang Chang et al. [1]. While Hang Chang et al. simply revealed its cytotoxicity against pancreatic cancer cells, our study further elucidated detailed mechanisms of action. Therefore, as the reviewer suggested, we modified the text on lines 21-22 and 61-63 and added the reference Hang Chang 2017 on line 61 of the introduction section.
- In the abstract, it is confusing to know that authors are stating matairesinol as a novel anticancer therapy, however, the overall manuscript suggest it as an adjuvant therapy in combination with 5-FU. Authors need to be sure about their saying; hence a better conclusive sentence is required.
Response: We appreciate and agree with the reviewer’s valuable comments. We revealed the anti-cancer effects of matairesinol in PANC-1 and MIA PaCa-2 cells and demonstrated the underlying mechanisms, including apoptosis induction, ROS generation, mitochondrial dysfunction, and calcium dysregulation. In addition, matairesinol showed synergistic effects with 5-FU against pancreatic cancer cells. Our results suggested that matairesinol can be used as a novel therapy against PDAC and as an adjuvant therapy. Therefore, according to the reviewer’s opinion, we modified the last sentence of the abstract into a better conclusive sentence.
- In the introduction section, Lines 61-62 ". However, the anticancer effect of matairesinol in the context of pancreatic cancer has not yet been elucidated." This line is wrong and misleading. Hang Chang et al 2017 have shown anti-cancer efficacy of these Lignons against pancreatic cancer cells. Please correct the sentence and include this reference in the text. Interestingly, this study lacks any mechanistic aspect which could be deemed as an asset to present manuscript.
Response: We appreciate and totally agree with the reviewer’s valuable comments. Following the reviewer’s suggestion, we corrected our manuscript and added the reference on lines 61-63.
- At the end of the Introduction section and Result section (2.6), how can authors be sure that Matairesinol exhibit synergistic effect but not an additive effect? More specifically authors should calculate synergistic or additive activity by calculating ISOBOLOGRAM etc. For more information read Ruo-Yue Huang et al 2019.
Response: We appreciate and totally agree with the reviewer’s valuable comments. As the reviewer mentioned, we calculated the bliss values based on the proliferation data using the Bliss independence model [2]. According to this model, the synergistic effect of a combination of two drugs was determined by the difference (D) between the observed (O) and the expected (E) inhibition of the combined treatment. E is calculated as follows: E = A+B-A•B, in which A and B are the inhibition rates of each drug. Therefore, positive and negative D indicate a synergistic and an antagonistic effect, respectively [3, 4]. Based on this calculation, the difference (D) is 0.015 and 0.076 in PANC-1 and MIA PaCa-2, respectively. Therefore, matairesinol exhibits a synergistic effect with 5-FU in pancreatic cancer cells.
- In the entire result section and figure legends, authors have not stated how many times each experiment was replicated. This is misleading and such missing information might ignite post-publication problems. Authors are requested to state how many times each individual experiment was replicated in the respective figure legend.
Response: We appreciate and totally agree with the reviewer’s valuable comments. We added the information in each figure legend according to the reviewer's suggestion.
- In the result section 2.1., authors stated that "Based on the observation that matairesinol could suppress the growth of PC cells, the protein level of PCNA was further confirmed in response to matairesinol treatment". I could not understand why suddenly PCNA levels were assessed. It could make readers confused. Authors need to describe why they only checked PCNA? Please do the needful.
Response: We appreciate the reviewer’s valuable comments. Proliferating cell nuclear antigen (PCNA) plays an important role in cancer proliferation, and its expression is involved in the biological activity of cancer cells [5]. Especially, pancreatic ductal adenocarcinoma (PDAC) exhibits a high expression of PCNA, which is deeply related to poor prognosis. Shanna J. Smith et al. have suggested that PCNA can be a novel therapeutic target for the treatment of PDAC [6]. According to the reviewer’s opinion, we added the information on PCNA with a reference on line 81 in the result section 2.1.
- In the result section 2.2., I could not understand the lines 105-106. Please reframe the lines to get the exact meaning.
Response: We appreciate the reviewer’s valuable comments and suggestions. According to the reviewer’s opinion, we modified the text on lines 105-106 to state the exact meaning.
- In the result section 2.2., authors stated in lines 108-109 "In particular, in response to 80 µM of matairesinol the relative late apoptosis was increased by up to 196% in PANC-1 (Figure 2A) and by 261% in MIA PaCa-2 cells (Figure 2B)". I could not understand. How late apoptosis is calculated as 196% & 261% in the bar graph, whereas the FACS based dot plot shows late apoptosis as 26.5% & 10.4%? Authors please explain or simplify to what is shown in FACS data. At the moment bar graph are highly misleading.
Response: We appreciate the reviewer’s valuable comments and suggestions on our manuscript. We expressed the “relative” fold change of apoptotic cells on the bar graph. We showed relative values rather than absolute values on the bar graph. Illustrated in Figures 2A and 2B, the late apoptotic cells at 0 µM in PANC-1 and MIA PaCa-2, which are the criteria, are 12.5% and 4.0%, respectively, and at 80 µM, they are 26.5% and 10.4%, respectively. In this way, these absolute values do not show a big difference, but when converted into relative values, the differences occur, as shown in the bar graph. However, the bar graph's purpose is to demonstrate the overall aspect, and the tendency is also consistent.
- In the figure 3A, 3B, 3C, 3D, FACS data in % does not correspond to the bar graph. It will be highly appreciable if authors could maintain uniformity between the FACS data and bar graph data.
Response: We appreciate the reviewer’s valuable comments and suggestions on our manuscript. Similar to the Annexin V and PI staining data, we expressed the “relative” fold changes induced by matairesinol on each bar graph compared with 0 µM. We showed relative values rather than absolute values on the bar graph. For a more precise expression, we added the term “relative” on the y-axis of each bar graph.
- In section 2.3., line 144-145, authors stated "Treatment with 80 µM matairesinol increased cytosolic calcium levels by 280% and 551% in both PC cell lines" (Figure 3C, 3D). Again there is same problem. How authors are calculating 280 or 551%? Whereas, FACS data limit cut-off is only 100%.
Response: We appreciate the reviewer’s valuable comments and suggestions on our manuscript. Similar to the previous FACS data, we expressed the “relative” fold changes induced by matairesinol on each bar graph compared with 0 µM. We showed relative values rather than absolute values on the bar graph. We calculated the relative cytosolic calcium levels as: (The average of calcium levels at 80 μM (=33.9 %))/(The average of calcium levels at 0 μM (=12.1 %))×100 (%)≈280(%). All bar graphs of FACS data are also represented in the same way.
- In section 2.4., authors have not justified the need for assessing only VEGF or FOXM1 or PLAU. Please specify the importance of assessing only these genes.
Response: We appreciate the reviewer’s valuable comments and suggestions on our manuscript. As the reviewer mentioned, we assessed only the expression levels of FOXM1, VEGF, PLAU, and MMP1 related to cell migration. Many reports have suggested that the high expression of these genes contributes to tumor invasion and poor prognosis. For this reason, these genes can be a target for the treatment of PDAC [7-10]. Of course, according to the reviewer’s opinion, the expression of other EMT-related genes, such as CDH1 (E-cadherin) and CDH2 (N-cadherin), should be confirmed, but this cannot be performed due to time constraints. It takes more than 3 weeks to purchase the primers and conduct these experiments. In a follow-up study regarding EMT, we will plan experiments involving other EMT-related genes.
- In section 2.6., title states "Synergistic effects of matairesinol….. ". However, the description states " First, matairesinol (80 µM) combined with 5-FU (20 µM) exhibited an additive antiproliferative effect ", This is confusing. Either it should be synergistic or it should be additive. Therefore, authors need to ascertain synergistic or additive activity by using ISOBOLOGRAM analysis. For more information read Ruo-Yue Huang et al 2019.
Response: We appreciate and totally agree with the reviewer’s valuable comments. As the reviewer mentioned, we proved that matairesinol exhibits a synergistic effect with 5-FU based on a Bliss independence model. According to the reviewer’s suggestion, we modified the text on line 227 of section 2.6.
- Bar graphs in Figure 6B-6F needs to be corrected based on the points raised in comment 8 and 9. Please do correct the bar graph in matching with FACS data.
Response: We appreciate the reviewer’s valuable comments and suggestions on our manuscript. Similar to the previous FACS data, we expressed the “relative” fold changes induced by matairesinol, 5-FU, and a combination of both chemicals on each bar graph compared with the control. However, the bar graph showed relative values rather than absolute values. For a more precise expression, we added the term “relative” on the y-axis of each bar graph.
- Figure 7 needs to be corrected based on the ISOBOLOGRAM analysis, which will suggest Synergistic or Additive effect. Alternatively, authors could remove Big Red Arrow pointing inside the cell towards PCNA along with 5-FU structure. Since the soul of the manuscript is based on the importance of matairseinol, I would suggest to only depicts the mechanism of matairesinol.
Response: We appreciate and agree with the reviewer’s valuable comments. According to the reviewer’s opinion, we modified figure 7 by removing the red arrow along with the structure of 5-FU.
- In the figure 7, what does CELL PROGRESSION in rectangular box means? Does it means cell proliferation or else? Please correct.
Response: We appreciate and agree with the reviewer’s valuable comments. According to the reviewer’s opinion, we modified figure 7 by correcting the text in the box from cell progression to cell proliferation.
16 . In the figure 7, Why only PCNA is in nucleus and not other genes (VEGFC, MMP1, FOXM1, PLAU). Manuscript does not show these genes or proteins to shuttle out of nucleus. So, nucleus size should be enhanced and encircling all these 4 genes.
Response: We appreciate and agree with the reviewer’s valuable comments. According to the reviewer’s opinion, we modified figure 7 to enlarge the nucleus and include all these 4 genes.
- In the figure 7, events such as Apoptosis, migration, cell proliferation should be shifted out of the cell structure as all these events are extracellular events.
Response: We appreciate the reviewer’s valuable comments. According to the reviewer’s opinion, we modified figure 7 by shifting the extracellular events into out of the cell structures.
- Lines 328-338, needs to be supplement with more references, since few drugs such as Moxifloxacin and ciprofloxacin has shown their efficacy against pancreatic cancer cells by enhancing pERK levels and decreasing pAKT levels (Yadav V et al 2015). More such examples are asiatic acid, pemetrexed, lauryl gallat Taxol etc.
Response: We appreciate the reviewer’s valuable comments. According to the reviewer’s suggestions, we added more references for lines 338-342 of the manuscript.
- Section 4.11., authors did not state the method or software to quantify the gap distance for invasion analysis. Please mention.
Response: We appreciate the reviewer’s valuable comments. According to the reviewer’s suggestion, we added the information about the method for calculating the gap distance for invasion analysis in section 4.11.
- Since in all the figures authors have used one-way ANOVA, it is important to mention the posthoc multiple comparison test. Please do the needfull for each figure legend.
Response: We appreciate the reviewer’s valuable comments. According to the reviewer’s suggestion, we added the information about the post hoc test in section 4.14. and each figure legend.
Minor Comments:
- In the abstract, authors stated in line 29-30 "Therefore, these results demonstrate ……context of PDAC as a novel anticancer therapy". This sentence is grammatically incorrect. Please reframe.
Response: We appreciate the reviewer’s valuable comments. According to the reviewer’s suggestion, we modified the text on lines 29-30.
- In section 2.6., Line 216-217 "................... and standard anticancer drugs against PDAC and 5-FU" are grammatically incorrect. Please rectify.
Response: We appreciate the reviewer’s valuable comments. According to the reviewer’s suggestion, we modified the text on lines 225-226.
- In the figure 7., small triangles showing the up or down regulation should also be placed for AKT and MAPK pathway proteins.
Response: We appreciate the reviewer’s valuable comments and added small triangles showing the up or down regulations in AKT, JNK, ERK1/2, and P38 proteins according to the reviewer's suggestion.
- In the figure 7., decrease in the spheroid formation by matairesinol is missing. Since it is an extracellular event, it should be outside the cell structure.
Response: We appreciate the reviewer’s valuable comments. According to the reviewer’s suggestion, we added the box out of cell structure, where decreased spheroid formation by matairesinol occurs.
- Please correct the line 275-276 of discussion section. I could not understand.
Response: We appreciate the reviewer’s valuable comments. According to the reviewer’s opinion, we modified our manuscript more precisely.
- In the section 4.12., please mention the amount of RNA (micro or nanogram) used for cDNA synthesis.
Response: We appreciate the reviewer’s valuable comments. According to the reviewer’s opinion, we added the information about the amount of RNA used for cDNA synthesis in section 4.12.
[Reference]
- Chang, H.; Wang, Y.; Gao, X.; Song, Z.; Awale, S.; Han, N.; Liu, Z.; Yin, J., Lignans from the root of Wikstroemia indica and their cytotoxic activity against PANC-1 human pancreatic cancer cells. Fitoterapia 2017, 121, 31-37.
- Liu, Q.; Yin, X. F.; Languino, L. R.; Altieri, D. C., Evaluation of Drug Combination Effect Using a Bliss Independence Dose-Response Surface Model. Stat Biopharm Res 2018, 10, (2), 112-122.
- Foucquier, J.; Guedj, M., Analysis of drug combinations: current methodological landscape (vol 3, e00149, 2015). Pharmacol Res Perspe 2019, 7, (6).
- Kong, W. B.; Sender, S.; Taher, L.; Villa-Perez, S.; Ma, Y. X.; Sekora, A.; Ruetgen, B. C.; Brenig, B.; Beck, J.; Schuetz, E.; Junghanss, C.; Nolte, I.; Escobar, H. M., BTK and PI3K Inhibitors Reveal Synergistic Inhibitory Anti-Tumoral Effects in Canine Diffuse Large B-Cell Lymphoma Cells. International Journal of Molecular Sciences 2021, 22, (23).
- Ye, X. L.; Ling, B.; Xu, H. R.; Li, G. Q.; Zhao, X. G.; Xu, J. Y.; Liu, J.; Liu, L. G., Clinical significance of high expression of proliferating cell nuclear antigen in non-small cell lung cancer. Medicine 2020, 99, (16).
- Smith, S. J.; Li, C. M.; Lingeman, R. G.; Hickey, R. J.; Liu, Y. L.; Malkas, L. H.; Raoof, M., Molecular Targeting of Cancer-Associated PCNA Interactions in Pancreatic Ductal Adenocarcinoma Using a Cell-Penetrating Peptide. Mol Ther-Oncolytics 2020, 17, 250-256.
- Wierstra, I., FOXM1 (Forkhead box M1) in tumorigenesis: overexpression in human cancer, implication in tumorigenesis, oncogenic functions, tumor-suppressive properties, and target of anticancer therapy. Adv Cancer Res 2013, 119, 191-419.
- Longo, V.; Brunetti, O.; Gnoni, A.; Cascinu, S.; Gasparini, G.; Lorusso, V.; Ribatti, D.; Silvestris, N., Angiogenesis in pancreatic ductal adenocarcinoma: A controversial issue. Oncotarget 2016, 7, (36), 58649-58658.
- Wu, M. Y.; Shen, M.; Xu, M. D.; Yu, Z. Y.; Tao, M., FOLFIRINOX regulated tumor immune microenvironment to extend the survival of patients with resectable pancreatic ductal adenocarcinoma. Gland Surg 2020, 9, (6), 2125-2135.
- Ito, T.; Ito, M.; Shiozawa, J.; Naito, S.; Kanematsu, T.; Sekine, I., Expression of the MMP-1 in human pancreatic carcinoma: relationship with prognostic factor. Mod Pathol 1999, 12, (7), 669-74.
Reviewer 2 Report
The research article submitted by Woonghee Lee et al, is impressive. However, I have some concerns related to the selection of experimental cell line models. Since the authors have mentioned they have used MiaPACA and PANC1, both are poorly differentiated cell lines.
The storyline could be more interesting if the authors had used at least one type of the well (CAPAN1/2 or HPAF-II), moderately (SU86.86/BxPc3 or HPAC) and poorly differentiated PDAC cell lines (PANC1, MiaPACA are good), with respect to normal pancreatic cell lines like HPDE or HPNE. This combination of experimental sets could give the through frame set of Matairesinol impact on overall PDAC grades.
The combination impact of Matairesinol and 5FU does not look that significant (Fig 6A), so what's the use of using the combination approach while Matairesinol itself is capable of inhibiting the growth of PDAC cellline?Author Response
[Comments and Suggestions for Authors]
[Reviewer 2]
The research article submitted by Woonghee Lee et al, is impressive. However, I have some concerns related to the selection of experimental cell line models. Since the authors have mentioned they have used MiaPACA and PANC1, both are poorly differentiated cell lines.
Response: We appreciate the reviewer’s valuable comments and suggestions regarding our manuscript. The reviewer mentioned that PANC-1 and MIA PaCa-2 cells are poorly differentiated PC cell lines [11]. Nevertheless, PANC-1 and MIA PaCa-2 are widely used cell lines for in vitro study of PDAC because they are easily accessible, reliable, and less tricky compared to the primary cultured cell [12, 13]. Moreover, in the tumorigenicity assay, tumors were well-formed when SCID mice received intraperitoneal injections of both tumor cells. Furthermore, their genetic and morphological characteristics are well studied [13], suggesting that both cell lines are good tools for in vitro study of PDAC.
The storyline could be more interesting if the authors had used at least one type of the well (CAPAN1/2 or HPAF-II), moderately (SU86.86/BxPc3 or HPAC) and poorly differentiated PDAC cell lines (PANC1, MiaPACA are good), with respect to normal pancreatic cell lines like HPDE or HPNE. This combination of experimental sets could give the through frame set of Matairesinol impact on overall PDAC grades.
Response: We appreciate the reviewer’s valuable comments and agree with the reviewer’s opinion. We wished to check the effects of matairesinol in other types of PDAC cell lines following your comments, but this could not be possible because of time constraints. It takes more than 3 months to purchase the cell lines and conduct these experiments. Therefore, in a follow-up study regarding PDAC, we will perform experiments with various pancreatic cancer cell lines, including well, moderately, and poorly differentiated PDAC cells and normal pancreatic cell lines.
The combination impact of Matairesinol and 5FU does not look that significant (Fig 6A), so what's the use of using the combination approach while Matairesinol itself is capable of inhibiting the growth of PDAC cellline?
Response: We appreciate the reviewer’s valuable comments. As the reviewer mentioned, matairesinol itself has shown significant inhibitory effects on the proliferation of both pancreatic cancer cell lines. However, patients with PDAC exhibit chemoresistance to the conventional anti-cancer drug 5-FU, which results in treatment failure in PDAC [14]. Moreover, Figure 6B-6F shows that the combination of matairesinol and 5-FU is more effective than the matairesinol treatment alone. Therefore, we confirmed the synergistic effects of matairesinol with 5-FU.
[Reference]
- Chang, H.; Wang, Y.; Gao, X.; Song, Z.; Awale, S.; Han, N.; Liu, Z.; Yin, J., Lignans from the root of Wikstroemia indica and their cytotoxic activity against PANC-1 human pancreatic cancer cells. Fitoterapia 2017, 121, 31-37.
- Liu, Q.; Yin, X. F.; Languino, L. R.; Altieri, D. C., Evaluation of Drug Combination Effect Using a Bliss Independence Dose-Response Surface Model. Stat Biopharm Res 2018, 10, (2), 112-122.
- Foucquier, J.; Guedj, M., Analysis of drug combinations: current methodological landscape (vol 3, e00149, 2015). Pharmacol Res Perspe 2019, 7, (6).
- Kong, W. B.; Sender, S.; Taher, L.; Villa-Perez, S.; Ma, Y. X.; Sekora, A.; Ruetgen, B. C.; Brenig, B.; Beck, J.; Schuetz, E.; Junghanss, C.; Nolte, I.; Escobar, H. M., BTK and PI3K Inhibitors Reveal Synergistic Inhibitory Anti-Tumoral Effects in Canine Diffuse Large B-Cell Lymphoma Cells. International Journal of Molecular Sciences 2021, 22, (23).
- Ye, X. L.; Ling, B.; Xu, H. R.; Li, G. Q.; Zhao, X. G.; Xu, J. Y.; Liu, J.; Liu, L. G., Clinical significance of high expression of proliferating cell nuclear antigen in non-small cell lung cancer. Medicine 2020, 99, (16).
- Smith, S. J.; Li, C. M.; Lingeman, R. G.; Hickey, R. J.; Liu, Y. L.; Malkas, L. H.; Raoof, M., Molecular Targeting of Cancer-Associated PCNA Interactions in Pancreatic Ductal Adenocarcinoma Using a Cell-Penetrating Peptide. Mol Ther-Oncolytics 2020, 17, 250-256.
- Wierstra, I., FOXM1 (Forkhead box M1) in tumorigenesis: overexpression in human cancer, implication in tumorigenesis, oncogenic functions, tumor-suppressive properties, and target of anticancer therapy. Adv Cancer Res 2013, 119, 191-419.
- Longo, V.; Brunetti, O.; Gnoni, A.; Cascinu, S.; Gasparini, G.; Lorusso, V.; Ribatti, D.; Silvestris, N., Angiogenesis in pancreatic ductal adenocarcinoma: A controversial issue. Oncotarget 2016, 7, (36), 58649-58658.
- Wu, M. Y.; Shen, M.; Xu, M. D.; Yu, Z. Y.; Tao, M., FOLFIRINOX regulated tumor immune microenvironment to extend the survival of patients with resectable pancreatic ductal adenocarcinoma. Gland Surg 2020, 9, (6), 2125-2135.
- Ito, T.; Ito, M.; Shiozawa, J.; Naito, S.; Kanematsu, T.; Sekine, I., Expression of the MMP-1 in human pancreatic carcinoma: relationship with prognostic factor. Mod Pathol 1999, 12, (7), 669-74.
- Deer, E. L.; Gonzalez-Hernandez, J.; Coursen, J. D.; Shea, J. E.; Ngatia, J.; Scaife, C. L.; Firpo, M. A.; Mulvihill, S. J., Phenotype and genotype of pancreatic cancer cell lines. Pancreas 2010, 39, (4), 425-35.
- Gradiz, R.; Silva, H. C.; Carvalho, L.; Botelho, M. F.; Mota-Pinto, A., MIA PaCa-2 and PANC-1 - pancreas ductal adenocarcinoma cell lines with neuroendocrine differentiation and somatostatin receptors. Sci Rep 2016, 6, 21648.
- Shen, Y.; Pu, K.; Zheng, K.; Ma, X.; Qin, J.; Jiang, L.; Li, J., Differentially Expressed microRNAs in MIA PaCa-2 and PANC-1 Pancreas Ductal Adenocarcinoma Cell Lines are Involved in Cancer Stem Cell Regulation. Int J Mol Sci 2019, 20, (18).
- Shi, X.; Liu, S.; Kleeff, J.; Friess, H.; Büchler, M. W., Acquired resistance of pancreatic cancer cells towards 5-Fluorouracil and gemcitabine is associated with altered expression of apoptosis-regulating genes. Oncology 2002, 62, (4), 354-62.
Round 2
Reviewer 1 Report
I would like to appreciate authors for successfully answering all my concerns.
I have a minor comment:
Since authors have used "Bliss independence model" to estimate the synergistic effect of Matairesinol, it is really important to mention this piece of information somewhere in the text with the citation of reference related to technique. This information will be helpful for researchers in future.
Congratulations.
Author Response
Reviewer 1:
I would like to appreciate authors for successfully answering all my concerns.
I have a minor comment:
Since authors have used "Bliss independence model" to estimate the synergistic effect of Matairesinol, it is really important to mention this piece of information somewhere in the text with the citation of reference related to technique. This information will be helpful for researchers in future. Congratulations.
Response: We appreciate the reviewer’s valuable comments again, and totally agree with the reviewer’s suggestion. According to the reviewer’s opinion, we added the information in section 4.14. of our manuscript.
Reviewer 2 Report
Thanks for the responding, the comments arise by me. Still, i feel that at least the whole experiment should be in comparison to normal pancreatic cell lines like HPNE. HPNE is very easy to grow in routinely available cell culture media; no need to grow this cell line in special conditions.
Believe me, its very important to give the strength of your experiment and to justify your whole hypothesis.
Author Response
Thanks for the responding, the comments arise by me. Still, I feel that at least the whole experiment should be in comparison to normal pancreatic cell lines like HPNE. HPNE is very easy to grow in routinely available cell culture media; no need to grow this cell line in special conditions.
Believe me, its very important to give the strength of your experiment and to justify your whole hypothesis.
Response: We appreciate the reviewer’s valuable comments again, and totally agree with the reviewer’s suggestion. The reviewer’s suggestion is very reasonable, and additional experiments with normal pancreatic cell line, HPNE should be conducted. However, we are currently facing some difficulties in conducting additional experiments with HPNE right now. First, to get the normal pancreatic cell, hTERT-HPNE from ATCC, our lab must obtain approval from our government. This is because our lab is not yet registered as a LMO facility. This procedure is very complicated and takes more than 2 weeks. We can order the cell line only after the approval from government. Moreover, currently, COVID-19 has not been completely resolved, and it is quite difficult to bring in cell lines from abroad due to the re-proliferation and the rapid increase of COVID-19. When we asked ATCC, it said it would take at least 6 weeks. In addition, it also requires minimum 2 weeks to conduct the experiments. Therefore, we need at least 10 weeks to conduct the additional experiment. Please kindly understand our circumstances. Of course, we deeply agree with the reviewer’s comment that additional experiments with HPNE cell line can justify our hypothesis. However, there is not enough time to order cell lines and perform further experiments following appropriate comments by the reviewer. Therefore, in a follow-up study, we will make sure to add an HPNE cell experiment to argue our hypothesis more convincingly.